# Simple and Autonomous Sleep Signal Processing System for the Detection of Obstructive Sleep Apneas

**DOI:** 10.3390/ijerph19116934

**Published:** 2022-06-06

**Authors:** William D. Moscoso-Barrera, Elena Urrestarazu, Manuel Alegre, Alejandro Horrillo-Maysonnial, Luis Fernando Urrea, Luis Mauricio Agudelo-Otalora, Luis F. Giraldo-Cadavid, Secundino Fernández, Javier Burguete

**Affiliations:** 1Department of Physics and Applied Mathematics, Universidad de Navarra, 31009 Pamplona, Spain; lfurrea@unav.es; 2School of Engineering, Universidad de La Sabana, Chía 25001, Colombia; mauricio.agudelo@unisabana.edu.co; 3School of Engineering and Basic Sciences, Universidad Central, Bogotá 110311, Colombia; 4Neurophysiology and Sleep Medicine, Clínica Universidad de Navarra, 31009 Pamplona, Spain; eurrestara@unav.es (E.U.); malegre@unav.es (M.A.); ahorrillo@unav.es (A.H.-M.); 5Epidemiology and Biostatistics, Internal Medicine, Universidad de La Sabana, Chía 250001, Colombia; luisf.giraldo@unisabana.edu.co; 6Department of Otorhinolaryngology, Clínica Universidad de Navarra, 31009 Pamplona, Spain; sfgonzalez@unav.es

**Keywords:** obstructive sleep apnea, sleep signal processing system, detection of apneas, signal processing, polysomnography, electrical stimulation, apnea, hypopnea

## Abstract

Obstructive sleep apnea (OSA) is a common sleep disorder characterized by repetitive upper airway obstruction, intermittent hypoxemia, and recurrent awakenings during sleep. The most used treatment for this syndrome is a device that generates a positive airway pressure—Continuous Positive Airway Pressure (CPAP), but it works continuously, whether or not there is apnea. An alternative consists on systems that detect apnea episodes and produce a stimulus that eliminates them. Article focuses on the development of a simple and autonomous processing system for the detection of obstructive sleep apneas, using polysomnography (PSG) signals: electroencephalography (EEG), electromyography (EMG), respiratory effort (RE), respiratory flow (RF), and oxygen saturation (SO_2_). The system is evaluated using, as a gold standard, 20 PSG tests labeled by sleep experts and it performs two analyses. A first analysis detects awake/sleep stages and is based on the accumulated amplitude in a channel-dependent frequency range, according to the criteria of the American Academy of Sleep Medicine (AASM). The second analysis detects hypopneas and apneas, based on analysis of the breathing cycle and oxygen saturation. The results show a good estimation of sleep events, where for 75% of the cases of patients analyzed it is possible to determine the awake/asleep states with an effectiveness of >92% and apneas and hypopneas with an effectiveness of >55%, through a simple processing system that could be implemented in an electronic device to be used in possible OSA treatments.

## 1. Introduction

Obstructive sleep apnea (OSA) is a highly prevalent condition, affecting about 27% of the general population (ranging from 9% to 38%) [1,2], and it is associated with outcomes having a great impact on the morbimortality of the general population, such as high blood pressure [3], coronary artery disease [4,5], stroke [6,7], and sudden death [8].

Positive airway pressure (PAP, including Continuous Positive Airway Pressure, CPAP, and Automatic Positive Airway Pressure, APAP) is the OSA treatment with greatest support in the scientific evidence [9]. However, PAP adherence is low with a percentage of use as low as 36% [10], forcing researchers to look for alternatives.

Electrical stimulation (ES) with implanted devices of the airway dilating muscles has been effective in reducing obstructive episodes during sleep by up to 68% [11,12,13,14]. However, when such a stimulation is performed using an implantable electrical stimulator, the procedure is almost non-reversible and invasive.

Therefore, the development of an external non-invasive device providing electrical stimulation specifically to the submental region and [15,16] to the airway dilator muscles (geniohyoid, sternohyoid, and genioglossus) [17] to improve upper airway patency during obstructive sleep apnea events seems to be a promising therapeutic alternative for OSA [18,19]. An external stimulator would require a component detecting obstructive sleep apnea events that would trigger the activation of electrical stimulation of dilator muscles to restore the patency of the upper airway. Such a component should validly detect sleep and obstructive sleep apnea and hypopnea events.

This article is part of the development of an external stimulator in our group. The electronic device being developed is a compact, comfortable, and manageable system, whose requirements are to be a portable device, with low power consumption, a reduced number of wires, and a simple control interface for a laptop or a mobile application. The biomedical device is made up of three parts. The first part is the electronic instrumentation that receives physiological signals associated with respiration (respiratory flow, respiratory effort, oxygen saturation, among others) and sleep analysis (electroencephalography and electromyography). The second part is an external transcutaneous and intraoral electrical stimulator that sends electrical signals with biphasic waves. Parameters, such as amplitude and frequency, can be configured to these waves. The third part is a signal processing algorithm that extracts signals from the electronic instrumentation, analyzes them, and then sends the results to the electrical stimulator control system.

This article is focused on the development of this algorithm, i.e., an autonomous and simple sleep signal processing system to detect sleep and obstructive sleep apnea and hypopnea events and we assessed its accuracy on full overnight polysomnograms read by sleep medicine experts in a university hospital sleep laboratory. The goal of this study is to determine a minimalist algorithm that can effectively detect when hypopnea or apnea appears in patients suffering from Obstructive Sleep Apnea Syndrome (OSA).

## 2. Materials and Methods

These patients were retrospectively selected from the database available at the Sleep Unit of the Neurophysiology Service of the Clínica Universidad de Navarra (CUN), Spain. The inclusion criteria were an age of 20 years or older who had undergone a PSG study at the CUN with an Apnea-Hypopnea Index (AHI) rate > 5. The exclusion criteria did not depend on the patient, but on the data series: artifacts cannot affect more than 55% of the total time. More than 200 records met these criteria, and a set of 20 were randomly selected.

The Ethics Committee of the University of Navarra verified that this work had no ethical concerns and was conducted in accordance with good clinical practices, the Helsinki Declaration, and national regulations. All of the participants received oral and written explanations and provided written informed consent for the anonymous use of data for research.

### 2.1. Data

Recordings were obtained using the Harmonie 6.0 system (Stellate Inc., Montreal, Canada). Eighteen variables were recorded for each patient with a sample rate of 200 Hz: 7 of electroencephalography (EEG), 2 of electrooculography (EOG), 1 of oxygen saturation (SO_2_), 2 of respiratory Flow (RF1 and RF2), 2 of respiratory effort (RE1 and RE2), 1 of electrocardiography (ECG), 2 of electromyography (EMG1 and EMG2), and 1 of Heart Rate (HR). The signals were acquired in a period long enough (10 h) to cover the average period of sleep.

The locations of the EEG leads were C3, C4, CZ, F3, F4, O1, and O2; the EMG leads are located on the chin. The respiratory flow sensors (pressure and temperature) measure the flow of air through a nasal cannula. Respiratory effort sensors consist of bands located in the thorax and abdomen with piezoelectric sensors. The electrodes and sensors were placed according to the recommendations of the American Academy of Sleep Medicine (AASM) Sleep Manual [20].

A sleep expert analyzed the recorded data and classified the sleep stages (awake, N1, N2, N3, and REM) and the presence of sleep apnea. Sleep stage classification was performed following the current AASM criteria, and apneas were scored using the standard recommended AASM scoring criteria (version 2.3) [21]. These sleep scorings were considered as reference standards to validate the signal processing system developed in this research.

### 2.2. Signal Analysis

Several tests were carried out using all the PSG signals, through different data sets. Eventually, only the essential signals were retained for the realization of a simple algorithm to detect the status of awake/asleep and sleep apnea.

Although we carried out tests with all EEG channels, we verified that the electrodes located in the occipital, frontal, and central parts of the head presented a greater change in the brain activity. This is in agreement with the common practice of sleep experts, that mainly use these 6 EEG channels for the analysis of sleep stages (occipital: O1, O2; frontal: F3, F4 and central: C3, C4). We retained only these 6 channels for the development of our algorithm.

The EMG channels located on the chin were also analyzed, which in several patients showed significant reduction in muscle activity associated with the sleep stages.

Although the electrodes and sensors were correctly placed, as a result of patient movement it is possible that some artifacts are created overtime on the signals due to sensor displacements that may have occurred and that could generate disconnections and poor-quality signals. Concerning artifacts found in the EEG and EMG channels associated with abrupt changes and large amplitudes, they are associated with possible electrode movements, deficient connections, scarce conductive gel use, or loss of reference electrode due to disconnection.

The polysomnography device has two flow sensors, one for temperature (via a thermistor) and one for pressure. For the algorithm, the second signal (pressure) of respiratory flow RF2 was taken, because this signal had less noise and artifacts and the amplitude drop was more remarkable when the subject was in hypopnea or apnea. Respiratory effort sensors placed around the chest (channel RE1) and abdomen (channel RE2) were also used, which provide useful information about the respiratory cycle and allow differentiating between obstructive apnea and central apnea. Regarding the oxygen saturation signal, the presence of hypopnea is inferred with a desaturation of more than 3 percentage points.

The remaining channels (EOG, ECG, and HR) were not used on the analysis of the present work.

The algorithm was developed in Matlab^®^ software (MathWorks, Natick, MA, USA) and it is divided into two parts: detection of awake/sleep stages and detection of sleep apnea.

#### 2.2.1. Detection of Awake/Sleep Stages

This analysis is centered on the hypothetical modification of the EEG and EMG signals when the patient changes between the “awake” and “sleep” stages. As referenced above, EEG alpha rhythm (8–13 Hz interval) disappears when a patient goes from awake to sleep. On the other side, EMG will vary because of changes in muscle activity, for example when the patient stops speaking or because of a reduction in mobility or due to muscle relaxation, which can be observed in the reduction in movements detected in the chin through EMG. EEG analysis will be effective only when the patient’s eyes are closed; if the patient’s eyes are open, the EMG analysis would detect sleep states.

The algorithm works with the digitized EEG and EMG raw signals recorded from each patient. An average signal is determined from all the EEG signals and is subtracted from each one of the EEG channels. Then, the signals are filtered with an equiripple (with low-amplitude 1 dB bandpass ripple and stop bands very close to the cutoff frequencies at 0.5 Hz) FIR-type [22] bandpass filter (bandpass frequencies 8 and 13 Hz) to retain only alpha waves activity [23]. The EMG signal is filtered (similar filter specs) in an interval of 24–45 Hz [24], for the detection of the reduction in muscular movements in the chin.

Each signal is divided into AASM epochs of 30-s. A Fast Fourier Transform (FFT) is computed for each signal to determine the presence of each frequency on the data [25,26]. For each epoch, the accumulated signals of the EEG and EMG signals were obtained in the filtered intervals.

A threshold value is selected to determine when the signal activity (either EEG or EMG) corresponds to awake or sleep stages. The accumulated amplitude of the filtered signals is compared against that threshold and so the corresponding epoch is marked by the algorithm as awake or sleep.

Expert stages N1, N2, N3, and REM were combined into a single stage: the expert sleep stage. This awake/sleep classification is compared with the classification obtained above for each specific threshold applied on the summations found with the FFT. The algorithm determines it as a success when there is a match between the state calculated and the annotation of the expert.

The threshold value is varied between the minimum and maximum accumulated value of the FFT Power. For each one of the threshold values, the algorithm determines the successes and the errors in each one of the epochs.

For each one of the signals of the EEG and EMG series and each threshold value, a confusion matrix was obtained. False negatives (FN) refer to the detection of a patient being awake when he/she is asleep and false positives (FP) refer to the detection of a patient being asleep when he/she is awake.

With the confusion matrices, ROC curves were constructed for each of the 6 EEG and 2 EMG signals of each patient. We could determine the optimum value that optimizes specificity vs. sensitivity from this ROC curve. However, we would like to recall that the application of this work would allow the development of an alternative treatment technology for apneas based on electrostimulation, which should act only when the user is sleeping. Given the above, the algorithm should obtain a low number of false positives, preventing the technology from sending stimulation pulses when awake.

We have included a different approach. For each signal, two thresholds have been determined: the first threshold is the ROC curve’s optimum (ROT) value, and the second value corresponds to the threshold (FPT) that produces a false positive rate below 10%. These thresholds are determined for each of the EEG and EMG signals in each of the patients. With the data from all the exams, a simple linear regression equation is obtained, relating false positive threshold FPT vs. the ROC curve ROT threshold. With this equation, we can improve the effectiveness of the algorithm.

Once this equation is determined, we can use it to recover the FPT thresholds for new patients once their ROT value is obtained from PSG analyses. Each patient will have a calibration stage in a hypothetical version of the device: a PSG is obtained and an expert classifies the data. From the corresponding ROC curve and the linear equation obtained above, a working threshold is obtained that ensures FP is below 10%

Through the linear equation, the algorithm obtains the FPT thresholds for each of the 6 EEG signals and 2 EMG signals for each exam analyzed. With these FPT values, the confusion matrices are found again.

Once the awake/sleep stages with each of the six EEG signals and the two EMG signals are found, these 8 binary signals (sleep = 1 and awake = 0) are combined by means of OR relationships. In the end, the algorithm searches for each subject for the best combination of EEG and EMG channels in detecting the awake/sleep states, which are compared with the scoring given by the sleep expert. Nine combinations were tested: (i) EEG signals, (ii) EMG signals, (iii) EEG + EMG signals, (iv) occipital EEG + EMG signals, (v) frontal EEG + EMG signals, (vi) central EEG + EMG signals, (vii) occipital EEG + frontal EEG + EMG signals, (viii) occipital EEG + central EEG + EMG signals, and (ix) frontal EEG + central EEG + EMG signals. The best combination corresponds to the one with the highest success rate (true positives + true negatives/number of epochs).

#### 2.2.2. Detection of Sleep Apnea

The second part of the algorithm works on the apnea detection analysis. According to the manual for scoring sleep and associated events of the AASM, different signals can be used to determine the existence of an apnea/hypoapnea event in adults: thermal or pressure flow sensors, PAP device flow, or another alternative sensor that can determine whether there is airflow. In this work, we applied the following rules for the detection of hypopneas and apneas [17]:An apnea is scored when there is a drop in peak signal excursion by ≥90% of the amplitude signal of the oronasal pressure sensor RF2 compared to previous epochs amplitudes and that remains for ≥10 s.A hypopnea is scored when there is a reduction between ≥30% and <90% of the amplitude signal of the oronasal pressure sensor RF2 compared to previous epochs, during ≥10 s in association with ≥3% arterial oxygen desaturation.

Artifacts found in respiratory channels are mainly associated with the displacement of the bands due to the nasal sensor and the movement of the body, which distorts the signal. Artifacts in the oximetry signal, in the same way, are generally associated with movement, which causes the optical sensors to work improperly, delivering an incorrect percentage of oxygen saturation.

Detection of artifacts in these signals was easy to identify. The artifacts saturated the acquisition system, so they had very large amplitudes, mostly reaching the limits of the analog-to-digital converter (ADC) of the corresponding channels of the PSG capture equipment. These amplitudes mean a saturated ADC, that is, the value is higher than the maximum or minimum of the converter, which is caused by an abrupt amplitude value generated by an artifact. The algorithm determines the range of data values and eliminates the lower 1% and the upper 1% of this range of values. In the case of an implementation in an electronic device, the values found in the highest amplitude of the ADC would be removed.

The previous process is carried out in each one of the epochs. Once an artifact was found in an epoch, the entire epoch was removed to avoid errors in the algorithm: The epochs where artifacts were found were labeled as “unusable epochs”.

The analysis of apneas is carried out from the amplitude of the breathing periods. The respiratory period of an average person lasts between 3 and 5 s. The algorithm analyzes for each epoch *i* the flow signal in sliding windows of 5 s (breathing period). On this window, the amplitude of the flow cycle is determined as the difference between the maximum and minimum values of the RF2 signal. This window is moved by steps of 0.5 s, so we recover an amplitude *A_i_*(*t*) with a resolution of 60 points per epoch (each 0.5 s). This amplitude *A_i_*(*t*) is compared to the baseline value, defined as the average of the amplitude on the previous epoch <*A_i_*_−1_*>*.

When there is a reduction greater than 90% of the amplitude (i.e., *A_i_*(*t*)/<*A_i_*_−1_> < 0.1) for more than 10 s, apnea is scored for this epoch. Then, the apneas are classified as central or obstructive depending on the behavior of the respiratory effort data from the chest and abdomen (RE1 and RE2). The data from both channels are split into epochs and the amplitude of the effort is determined in the same way as in the RF2 analysis with 5 s sliding windows.

When we observe that in an epoch where we detect an apnea there is a reduction in the ER1 and ER2 signals below 20% of the maximum value of the same epoch, and this reduction lasts for a period longer than 10 s, then this event is labelled as central apnea. Otherwise, it is marked as obstructive apnea. Two binary signals were created: central apnea and obstructive apnea with the calculation of each type of apnea for each epoch.

When there is a reduction in the flow amplitude *A_i_(t)* between 30% and 90% of the baseline amplitude, i.e., 0.1 < *A_i_*(*t*)/<*A_i_*_−1_*>* < 0.7, we look for a simultaneous reduction of 3% of the arterial saturation. This is obtained from the analysis of the SO_2_ signal: the difference between maximum and minimum values on an epoch is obtained. If this difference is larger than 3 percentage points and the condition for flow amplitude reduction is fulfilled, the algorithm determines that there is a hypoapnea event at that epoch.

The precision of the algorithm was obtained by calculating confusion matrices generated from the results of the processing algorithm and the scoring given by the sleep experts, where only the annotations of hypopneas and obstructive apneas were taken for the analysis. The sum of the true positives and the true negatives were taken and divided into the total epochs of each file. This value found in the analysis of the files of each subject was called success rate.

### 2.3. Statistical Analysis

The developed algorithm performs an analysis in the first part, classifying between the awake and sleep stages. The analysis was performed in Matlab^®^ software (MathWorks, Natick, MA, USA). A receiver operating characteristics (ROC) analysis was performed, obtaining the sensitivity and specificity of the awake/sleep stages compared to the sleep expert taken as the gold standard. The variables were calculated as follows:Sensitivity = true positive/(true positive + false negative)(1)
Specificity = true negative/(true negative + false positive)(2)

## 3. Results

The final group consisted of 20 subjects (51.4 ± 15.4 years) that were treated in the years 2017 and 2018. The characteristics of the subjects were: 16 men, 4 women, 7 with arterial hypertension (35%), 10 with hypercholesteremia or overweight (50%), 4 with asthma (20%), and 9 with previously diagnosed sleep apnea (45%).

The results of the developed algorithm are presented in two parts: detection of sleep-wakefulness states and apnea/hypopnea detection.

Figure 1 shows an example of an EEG signal in the sleeping and awake periods, with their respective FFT extracted. Frequency analysis (Figure 1b,d) shows the full spectrum in blue and the filtered signal between 8 and 13 Hz (alpha band brain activity) in orange.

For the detection of the sleep stage, a comparison was made with the awake/sleep classification recovered from the stages determined by the expert. Figure 2 shows the (a) FFTs Power accumulated in blue, (b) the expert’s annotations in red, and (c) the sleep stages given by the algorithm in magenta.

The distinction between the awake and sleep states was performed with the analysis of confusion matrices. The results of the confusion matrices (true positives, true negatives, false positives, and false negatives) for the calculated thresholds were plotted for each signal and patient. Table 1 shows an example of a confusion table for one of the EEG channels. The total number of “Awake” events that the algorithm generates is only 3.4% larger than the total number of “Awake” events indicated by the expert. On the other side, concerning the “Sleep” events, the algorithm detects 1.3% fewer events when compared with the expert. These differences are due to a total of 12 individual events.

Figure 3 shows the calculated ROC curves of the EEG signals for a single patient. Each curve is obtained from the confusion matrices of the calculated signal and the signal given by the expert determined for 100 different thresholds between the minimum and maximum values.

As explained above, and in order to increase the effectiveness, the developed algorithm generates a relationship between the optimal point of the ROC curves (ROT) and a point in which the number of false positives falls below 10% (FPT). Figure 4 shows the false positives in one of the analyzed patients. In this case, there are 348 epochs in the awake state and 872 epochs in the sleep state, labeled by the expert. The maximum threshold value (FPT) that produces a number of false positives (an erroneous “sleep” state) below 10% is stored. This value is signal dependent (i.e., different for each of the EEG channels). This specific patient (Figure 4) corresponds to the maximum threshold value with fewer than 35 false positives.

This relationship was found for each of the 6 EEG and 2 EMG signals in the 20 examinations analyzed. With the information about all the relationships found, the adjustment equation was generated through a linear regression of the normalized data between 0 and 1 (Figure 5).

Figure 5a shows the data for EEG signals. There is a clear correlation below 0.4, and the data become more scattered for higher thresholds. If the points with a ROC Curve normalized optimal threshold ROT/ROT_max_ above 0.6 were removed (these points correspond to series with a low AUC), the correlation R^2^ rises to 0.74, which indicates a good relationship with 85% of the thresholds analyzed.

In Figure 5b, we present the data for the EMG channels. A clear correlation is observed for all the thresholds analyzed, with R^2^ = 0.77.

As stated above, nine different combinations of the EEG and EMG signals were evaluated. The results concerning two specific combinations are presented in Figure 6. Figure 6a includes case (iii), where the whole set of EEG and EMG signals are combined. Figure 6b presents the data for the case with the highest success rate, case (iv), that uses only the occipital EEG signals combined with the EMG data.

The algorithm achieved an average accuracy—defined as (total number of epochs—false positives/total number of epochs—of 93.85% (95.38% median)) compared to the expert in detecting the sleeping stage. From these data, we can conclude that in this retrospective analysis of this cohort of 20 patients, the awake/sleep stage detection algorithm results were good when combination (iv) was used.

The second part of the algorithm performs hypopnea and apnea detection. First, the quality of the data and the RF2, SO_2_, ER1, and ER2 signals were analyzed. Several epochs presented artifacts or abrupt changes in the signal, which does not allow for a good analysis. This information was removed from the analysis, leaving between 55 and 85% valid information, regarding the epochs where the expert indicates that the person is sleeping.

The analysis of the behavior of the algorithm in the detection of apneas and hypopneas is analyzed in three aspects, shown in Figure 7.

False positives indicate the detection of hypopnea/apnea in a patient when the patient does not have hypopnea/apnea and false negatives are the detection of non-hypopnea/apnea when the patient has hypopnea/apnea. Because the technology proposed to be developed is activated only when hypopnea or apnea are detected, it is necessary that this factor (false positive) be reduced as much as possible. With this approach, we prevent the stimulation of the patient when there is no hypopnea/apnea, but for the same reason false negativity will increase and we can miss some hypopnea/apnea processes.

The false positives generated during the apnea calculation are shown in Figure 7a: all the patients have false positives lower than 9% (bigger outlier) with a median of 1.3%, so this demonstrates that this is an effective algorithm for apnea prognosis.

In the case of false positives in the detection of hypopneas (Figure 7b), the effectiveness of the algorithm decreases, showing a median of 4.46% and the 75th percentile of 10.05%. Two outliers are shown in the whisker diagram, and in those cases, the algorithm was not effective (FP around 22%). In this specific patient as well as the data that are above the 75th percentile, the algorithm detected SO_2_ data drops greater than 3% in various epochs. Nevertheless, in these periods the sleep expert did not label any apnea episode. Consequently, the analysis of the SO_2_ signal made a poor detection of a large number of hypopneas when correlated with the expert labels.

It should be noted, however, that the experts labeled hypoapneas as two different events: (i) when an event complies with the rules described above, at the beginning of Section 2.2.2 and (ii) when there is an arousal without desaturation. This last effect cannot be observed with our algorithm, and consequently the number of hypoapneas that our algorithm detects is different from the number of hypoapneas labeled by the expert.

If we assume a mean FP of approximately 7% with a person with an apnea/hypopnea index (AHI) greater than 30, this percentage corresponds to an error in detecting two hypopneas on average. The above implies two additional events in which the electrostimulator would act on the user per hour in a hypothetical implementation in a real device, which would be acceptable.

Figure 7c shows a whisker diagram with a median of 65.94%, with an average of 64.96% of the sum of apnea and hypopnea success. The success rate is larger than 55% in more than 75% of the patients, which indicates that the algorithm detects half of the actual hypopneas and apneas. Furthermore, 39.82% was the lowest result. Nevertheless, we would like to highlight that this success rate strongly depends on the quality of the data. Any improvement of the acquisition process will have an impact on improving these rates. The implementation of the algorithm shown could be carried out in a technology focused on sending electrical pulses to the airways when a person has apnea or hypopnea, with a device acting in 6 out 10 of the real events, which would allow an interesting treatment to be developed with a potential reduction to one half of the Apnea-Hypoapnea Index.

## 4. Discussion

We studied a new approach to detect apneas/hypoapneas that relies on an algorithm with two different parts: the first part detects whether the subject is asleep or awake, based on the analysis of EEG and EMG channels, and the second part is a simple analysis of the flow sensors and oximetry signals that detect apneas and hypoapneas.

Regarding the strengths and limitations of this study, our work is based on a reduced set of records that present an important number of artifacts. Any improvement in the quality of these data will improve the success rate of our algorithm. A clear advantage of our approach is that it relies on a simple analysis that can be easily implemented, with low computational requirements and low power consumption that makes this method suitable for wearable devices.

The algorithm proposed in this work for the classification of awake/asleep states presents an accuracy, using the best combination of EEG channels (using both Occipital electrodes) and two EMG channels, that ranges between 80.8% and 99.9%. On the other hand, it is a simple autonomous algorithm based on frequency analysis through FFT that can be easily implemented in an electronic device, compared to other approaches based, for example, on artificial intelligence algorithms.

Previous works by other groups differ on the number of sleep stages that are detected or the number of EEG Channels. There are considerable differences on the achieved success rates: using a single EEG channel, the success rate obtained is 83% in [27] and 91% in [28]. Depending on the number of sleep stages, a success rate of 77% is obtained when working with three sleep states in [29]; 90% when working with four sleep states in [30]; and 79% in [31], 81% in [32], and 74% in [33] when dealing with five sleep stages. Lastly, a 90.5% success rate is obtained when working with six sleep states in [34].

Studies using four EEG channels were also found [35] with an accuracy of 77.3% in a study of 243 subjects. Other works include other signals, such as EMG, for example in [36], with a success rate of 81% on a cohort of 49 subjects.

The data processing techniques are based on different approaches: Analysis with spectral edge frequency (SEF) in the 8–16 Hz frequency band [27], a state-space based sleep stages classification method [32], and the use of complex algorithms based on artificial intelligence: support vector machine (SVM) [28], deep neural network (DNN) [29], convolutional neural network [35], Neural-Network-Based Decision Tree [31,36], or random forest classifier [34].

A systematic review of 70 studies of wearable devices using different approaches for sleep staging was recently published [37]. Only a few of the analyzed devices presented a success rate close to the result presented in Figure 6. It is important to note that the 10% threshold used to optimize this classification can be modified so the number of false positives can be further decreased.

The detection algorithm proposed in this work for the awake/sleep stages could, from the analysis of the alpha brain wave and muscle activity (EMG) in the chin, have an average error of 4.4% (median 4.2%). A sleep period of approximately 8 h contains 960 epochs; in this algorithm, an error (where the patient was actually awake, but the algorithm detected him/her to be sleeping) could be generated in 40 epochs, that is, 20 min.

In the implementation of this algorithm in an electronic device, an initial calibration phase *per patient* is necessary to establish the thresholds to be applied to the signals. This calibration is based on performing a PSG examination, analyzed by an expert that indicates the epochs when the person is in the awake/sleep stages. With this information, the algorithm checks, case by case, the optimal values of the EEG and EMG channel thresholds that deliver the best detection of the sleep and awake states. Using the linear regression determined above, the algorithm will define the threshold values that produce less than 10% false positives. This set of values will be taken as a baseline for the device to begin to work.

Regarding the detection of apneas and hypopneas, several algorithms have been developed that differ on the selected biophysical signal and on the specific analysis tool used in the study. For example, an accuracy of 79% was obtained in [38], where the authors analyzed oximeter data through decision trees working with fragments of the SpO2 signal.

Another work was carried out by [39] with an accuracy of 79%. SpO2 and Photoplethysmography (PPG) signals were analyzed in segments of one minute, extracting characteristics in the time- and frequency domain and doing a pulse rate variability (PRV) analysis to identify all OSA events.

Other works analyzed nasal pressure, thermal sensors, and/or respiratory inductance plethysmography belts [40,41]. An accuracy of 96% was obtained in [42] using specialized PSG equipment (model: Alice LE, Philips Respironics). The algorithm presented in that work analyzes respiratory signals in 8-s segments with a Support Vector Machine (SVM) classifier. Some systems have also been proposed that use simple electronic systems for signal analysis [43] or diagnosis and treatment of OSA [44] based on patterns recognition: wavelet entropy, RMS value, and variance extracted from the respiratory effort signals to classify the presence or absence of apneas using machine learning with an accuracy of 82 ± 7%.

The processing system developed in this research obtained an accuracy detection of apneas and hypopneas at a minimum value of approximately 40%, which is a low value, related to what was obtained in previous works by other groups. However, despite the low percentage, the objective of this work was the development of a simple and minimalist algorithm that can be used for OSA treatment. With this success rate on the detection of apnea/hypo-apnea events, an implementation of this algorithm in an electronic device could reduce the AHI to one half the original value.

## 5. Conclusions

We have presented a simple algorithm that uses amplitude and time analysis to detect sleep/awake states and apnea or hypopnea events, using a minimal set of signals from polysomnography examinations. The algorithm needs a calibration stage, specific for each patient, with an analysis carried out by a sleep expert. This calibration determines different thresholds that, applied on the accumulated amplitudes of the filtered range of frequencies of the EEG and EMG channels, allows for the detection of whether the patient is awake or sleep, with a precision in the prognosis greater than 97% in 75% of the cases and 88% for the whole cohort of patients.

The analysis presented in this work uses six different EEG leads, but the number of EEG signals could be reduced to only two in a realization of the autonomous device: the occipital leads O1 and O2. In 17 of the 20 subjects evaluated, this was the best combination that increased the success rate to values above 90%. This would simplify a potential wearable device and consequently we could expect a reduced number of problems and artifacts when the patient uses the device at home without supervision.

For the detection of apneas or hypopneas, amplitude analyses were performed on the FR, RE, and SO_2_ signals. The success rates in the detection of hypopneas and apneas were greater than 55% in more than 75% of the patients analyzed with a mean rate of false positive below 4.3% in apneas and 7.09% in hypopneas. For example, in a person with an AHI greater than 30, the possible technology to be developed could act between 12 and 20 times per hour.

The implemented system can also have a self-learning process, with which the adjustment of the thresholds could be improved, increasingly improving the effectiveness in the detection of awake/sleep states and episodes of obstructive sleep apnea.

Due to the presence of artifacts, a handicap for this analysis that affected the success rates is the quality of the data series. An implementation of this algorithm in a real device can benefit from better electronic instrumentation that (i) prevents disconnections and (ii) reduces the displacement of the sensors.

We have shown that a simple algorithm with satisfactory success rates is achievable. This algorithm can be implemented in a simple electronic system with low requirements (both for power consumption and computing power) for the treatment of sleep apnea by means of an electrical stimulator controlled by polysomnography signals. Our results show that a software with these characteristics is feasible and can be implemented in an autonomous, simple, and low-cost system, composed of a simple microcontroller that allows carrying out amplitude calculation and frequency analysis operations.

## Figures and Tables

**Figure 1 ijerph-19-06934-f001:**
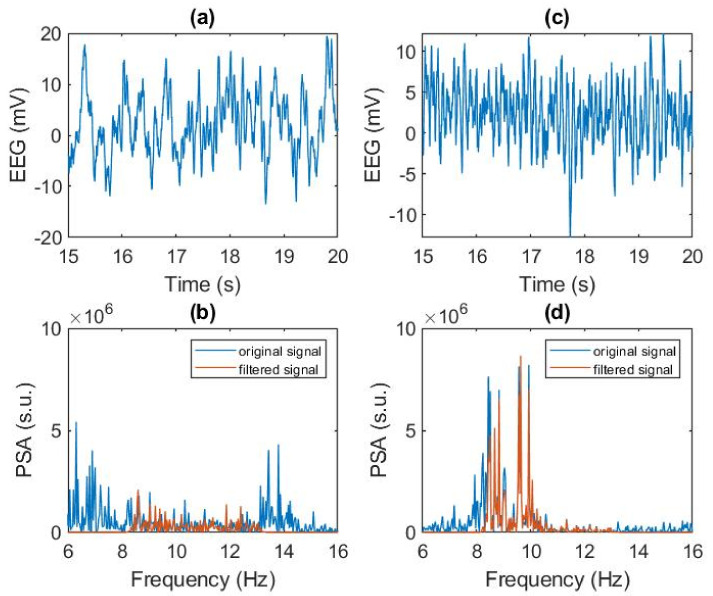
Real signals and FFT of the occipital EEG channel O1 in awake and sleep epoch. (**a**) Asleep original signal, (**b**) Frequency analysis of sleep signal, (**c**) Awake original signal, (**d**) Frequency analysis of awake signal.

**Figure 2 ijerph-19-06934-f002:**
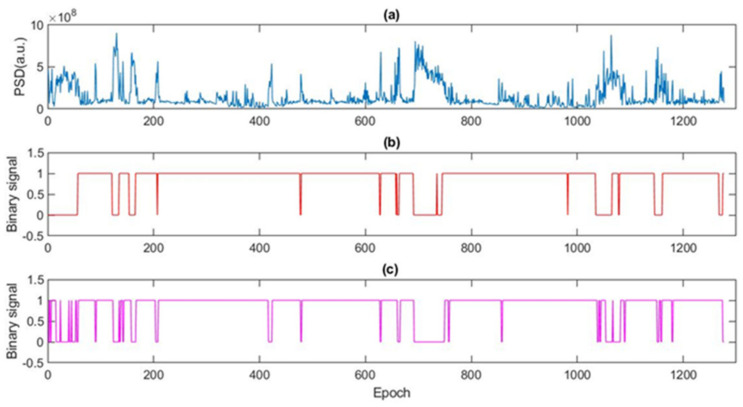
Accumulated signal of frequencies and states given by expert and algorithm. (**a**) Cumulative frequencies of the FFT signal of one of the EEG channels analyzed, (**b**) Binary signal of the awake/sleep stages given in annotations by the sleep expert, and (**c**) Binary signal of the generated awake/sleep stages by the signal processing algorithm.

**Figure 3 ijerph-19-06934-f003:**
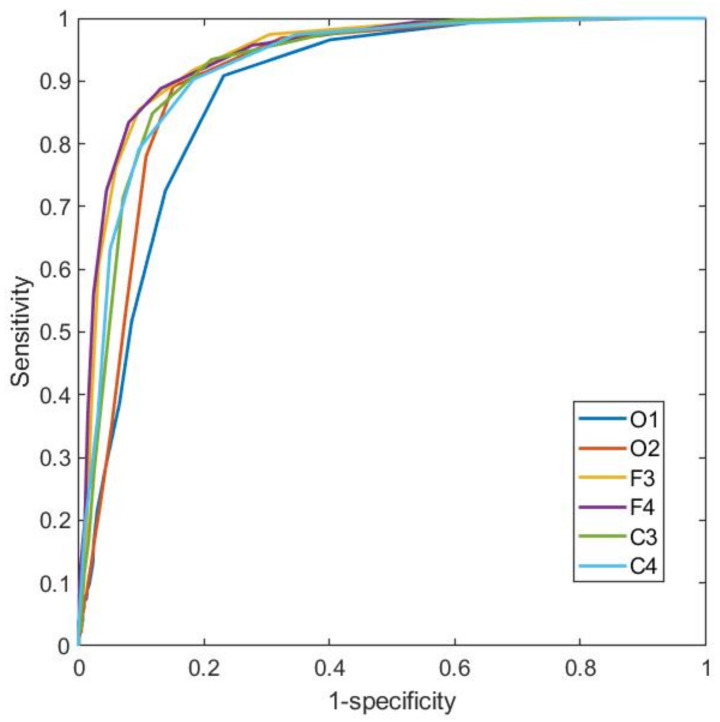
ROC curve EEG signals. The ROC curves of the 6 EEG channels taken for signal analysis are shown. All the channels show AUC greater than 0.8.

**Figure 4 ijerph-19-06934-f004:**
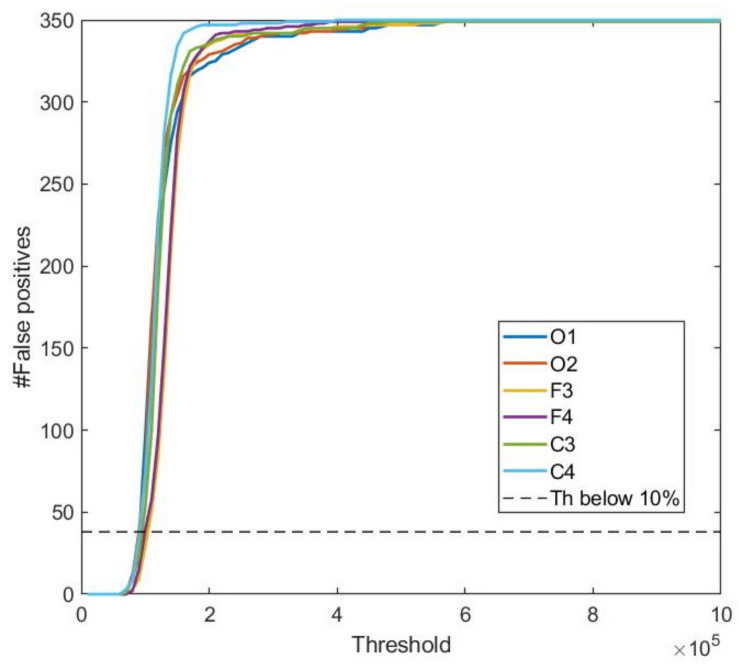
False positive EEG signals. The *X*-axis shows the thresholds evaluated in the detection of the awake/sleep stages in one of the subjects. The *Y*-axis shows the number of epochs with false positives.

**Figure 5 ijerph-19-06934-f005:**
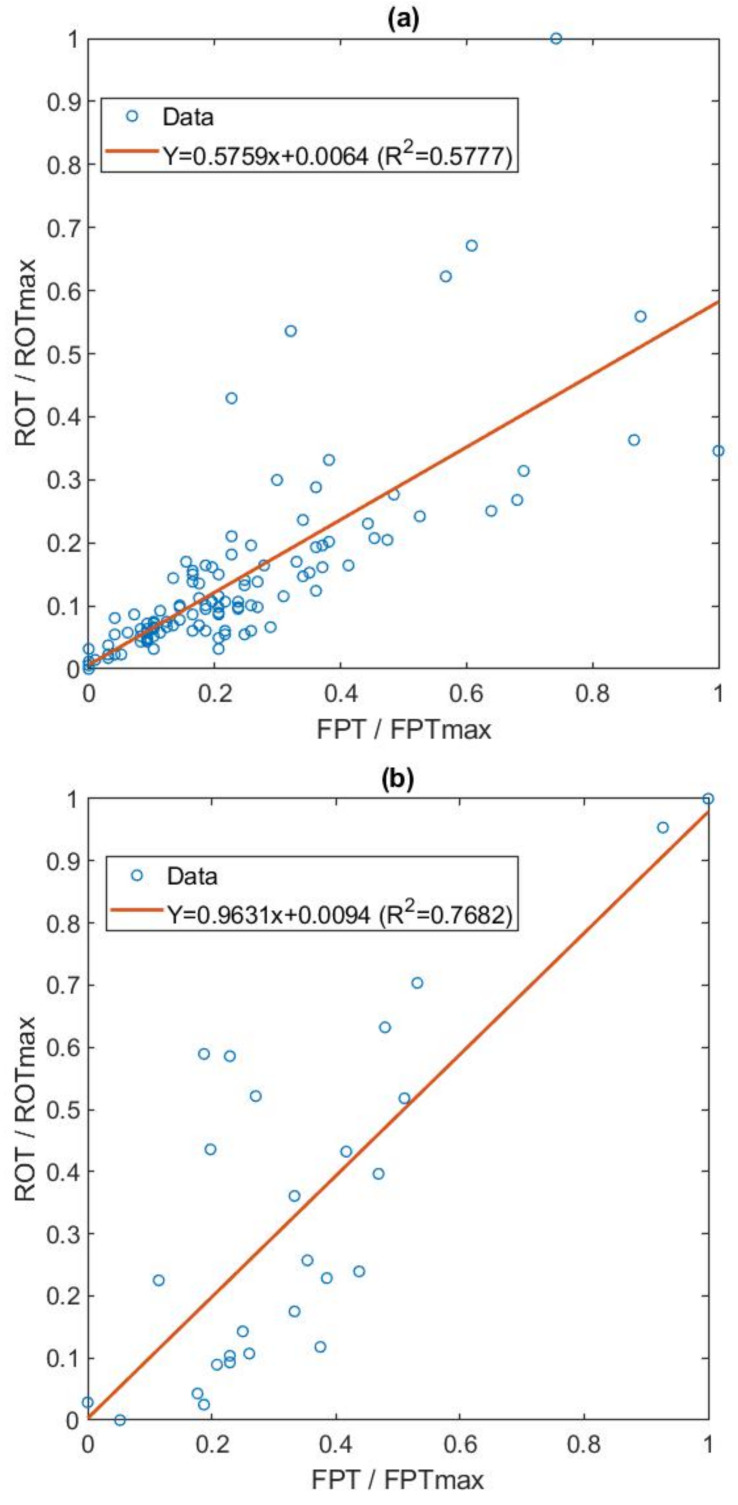
Relationship between ROC curve and 10% false positives. (**a**) Linear relationship given in all files and EEG channels of the 20 subjects and (**b**) Relationship given in all files and capture of chin EMGs in the 20 subjects.

**Figure 6 ijerph-19-06934-f006:**
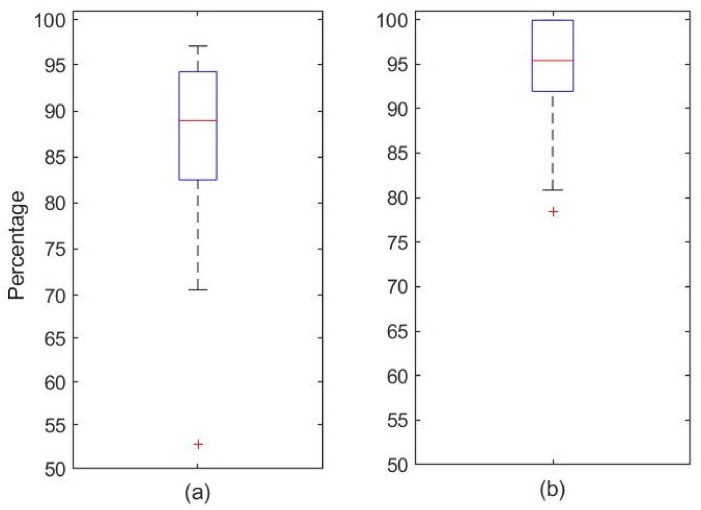
Success rate boxplot for awake/sleep detection on the evaluated subjects. (**a**) Case (iii): all EEG signals + EMG signals. Maximum, 97.04%; minimum, 70.57%; median, 88.93%; 25th percentile = 82.50%. (**b**) Case (iv): occipital EEG signals + EMG signals. Maximum, 99.91%; minimum, 80.84%; median, 95.38%; 25th percentile, 91.92%.

**Figure 7 ijerph-19-06934-f007:**
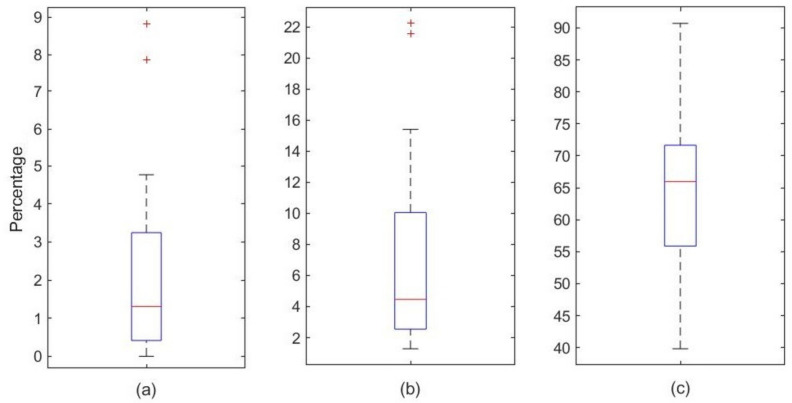
Boxplots of the apnea detection algorithm. (**a**) False positives of apneas, (**b**) False positives of hypopneas, and (**c**) Success rates in apneas + hypopneas.

**Table 1 ijerph-19-06934-t001:** EEG channel confusion table (F2) for a threshold of 0.1 × 10^5^. Each column corresponds to the real data (determined by the expert) and each row the algorithm determination by the Matlab program. (For these specific data, sensitivity = 0.83 and specificity = 0.92.)

Real (Expert)				
		Awake	Sleep	
**Algorithm**	**Awake**	292	70	362
	**Sleep**	58	802	860
		350	872	

## Data Availability

The data presented in this study are available on request from the corresponding author. The data are not publicly available due to legal reasons.

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
