# Peer review of "Simple and Autonomous Sleep Signal Processing System for the Detection of Obstructive Sleep Apneas"

_ijerph, 2022, doi:10.3390/ijerph19116934_

Round 1

Reviewer 1 Report

The authors adequately addressed the questions I raised. I recommend to accept the revised manuscript. 

Reviewer 2 Report

According to the reviewer’s comments, the authors have properly responded and corrected the manuscript. Now, this article could be accepted for publication after copyediting.

This manuscript is a resubmission of an earlier submission. The following is a list of the peer review reports and author responses from that submission.

Round 1

Reviewer 1 Report

I have reviewed the paper by Moscoso-Barrera et al. The authors took the topic of a simple and autonomous sleep signal processing system for the detection of obstructive sleep apneas. The following issues should be addressed:

Major comments:
- the inclusion and exclusion criteria should be stated
- how many patients from 2017-18 met these criteria
- We have 2022, why these patients were recruited 5years ago?
- There is a lack of proper description of performed statistical analysis
- Results and discussion should be in separate sections
- Please provide the strengths and limitations section

Minor comments:
- abbreviation should be introduced in brackets after the full name - please correct line 22
- all abbreviations should be used at least 2 times. The majority of abbreviation in the abstract is not used even once
- CPAP abbreviation is not introduced in the full text, just in manuscript

Author Response

Dear referee,

We would like to thank the referee for his/her careful reading of our manuscript. His/her comments have helped us to improve the readability of our paper, and to highlight the main points of our research.

We include answers to each one of the different comments. The original text of the referee is in blue, our answer appears in the following indented paragraph (in black)

Kind regards.

Reviewer 2 Report

The current manuscript aims to develop and evaluate novel signal processing system to detect obstructive sleep apneas and hypopneas. This system will be a part of the portable device that will automatically detect apnea and hypopnea events and send electrical signals to the hypoglossal dilating muscles to reduce obstructive events in patients with OSA.  This study is important and significant contribution to the development of new approaches and methods for obstructive sleep events detection and OSA treatment.  I have a few questions and suggestions that likely improve the current manuscript.

  1. In the Introduction and Conclusion sections, please discuss existing obstructive events detection approaches, their success rates and emphasize the novelty of your approach
  2. It is well known that obstructive events occur during sleep and these events are not present during wakefulness in OSA patients. What is the necessity of detecting of the sleep-wake stages for the detection of apneas and hypopneas?
  3. Line 443. Please replace “Figure 8C” with “Figure 7C”
  4. The success rate for obstructive apneas/hypopneas detection was 55% in 75% of the patients. Please report the success rate for the whole cohort of the patients. Please take into account the artifacts and report estimated success rate. Please compare the effectiveness of your method of obstructive events detection with other existing approaches.            

Author Response

Dear referee 

We would like to thank the referee for his/her careful reading of our manuscript. His/her comments have helped us to improve the readability of our paper, and to highlight the main points of our research.

We include answers to each one of the different comments. The original text of the referee is in blue, our answer appears in the following indented paragraph (in black)

Kind regards.

Reviewer 3 Report

This study aimed to determine a minimalist algorithm that can effectively detect when hypopnea or apnea appears in patients suffering from OSA. The subject and the initiative are interesting, but I have some concerns about the manuscript. 

Introduction

  • I consider more appropriate to move the aims of the study to the last paragraph of the introduction

Methods

  • According to the international literature, I consider the acronym OSA more appropriate to describe the obstructive sleep apnea syndrome. In the first paragraph of the manuscript authors used OSA and then in the methods opter for OAS.
  • Add the acronym for apnea/hypopnea index in the first mention
  • Complete the study setting by adding the country after “ Clínica Universidad de Navarra”
  • Inclusion/exclusion criteria?
  • Ethics statement should be addressed in the methods section
  • The sentence “The final group consisted on….” and the next paragraph should be moved to the Results section
  • I could not find/access the reference 17. Please provide DOI.
  • Statistical methods should be stated.

Results

  • Table 1: This table compares the results obtained by the algorithm versus the results obtained by the experts regarding sleep/awake state determination. The legend information is consistent with the columns but the rows are confusing to view. Which statistical method was used in this analysis? Please review this table.  
  • Figure 3 - ROC curve: What was the criterion for selecting this single patient in this figure?
  • Figure 8 is not available in the PDF/files provided.

Discussion

  • The authors have chosen to merge the Results and Discussion sections. However, there is no adequate discussion in light of the available literature. For this reason, the discussion section is very important for the reader to understand the rationale of the results in comparison to other available studies. Only one study (systematic review) was referenced to discuss the results.

Conclusions

  • The conclusion does not necessarily answer the purpose of the study. It seems that the goal of the paper was to determine the algorithm. Throughout the manuscript there was no mention of the feasibility of creating a device, as well as its complexities and costs.

Additional comments:

  • Authors need to prove that ethical review and approval were waived or to attach a document that proves the regularity of the research project according to local laws. Although it depends on national legislation, in most countries, ethics committees require approval for scientific studies, even with a retrospective design. The data were collected from Clínica Universidad de Navarra (Spain) but funded by Colombian bodies. I kindly ask the authors to clarify this question.
  • Additionally, authors state that  “Informed consent was obtained from all subjects involved in the study”. How was this process done, since the study had a retrospective design?
  • Data Availability Statement is applicable and should be stated by authors. 
  • Conflicts of Interest: Is there a patent process underway or an intention to commercialize this device/software? Do any of the authors market or provide services/products to sleep medicine related companies? This should be clearly stated by the authors as this may constitute a conflict of interest.
  • The purpose of the work is unclear. Throughout the manuscript the authors evoke three different goals:
  1. “The goal of this study was to determine a minimalist algorithm that can effectively detect when hypopnea or apnea appears in patients suffering from Obstructive Sleep Apnea Syndrome (OAS)”
  2.  “But we would like to recall that the aim of this work is to develop an alternative treatment technology for apneas based on Electrostimulation, which should act only when the user is sleeping” (Line 181)
  3. “The objective of the processing system shown was to obtain a simple algorithm that can be implemented in an electronic system for the treatment of sleep apnea by means of an electrical stimulator controlled by polysomnography signals” (Line 470)
  • The manuscript has important data and ideas. However, I suggest that the authors focus the text only on the objective of the work. In my opinion, mentioning the electrostimulator several times is confusing to the reader because it was not covered in this study.

Author Response

(The authors gave the same response as above.)

Round 2

Reviewer 3 Report

According to the response letter, apparently the authors made an effort to meet the reviewer's requirements. However, apparently they submitted an incorrect version of the manuscript in the submission system (maybe by mistake). I was unable to find the changes noted in the response letter. The manuscript provided is also not easily readable. Therefore, I request that the authors submit a revised version of the manuscript with the highlighted changes and not with Word tracked-changes. This will allow me to evaluate the revisions made by the authors and complete a new review round.
Sincerely,